# The Role of Cytokines and Chemokines in Shaping the Immune Microenvironment of Glioblastoma: Implications for Immunotherapy

**DOI:** 10.3390/cells10030607

**Published:** 2021-03-09

**Authors:** Erica C. F. Yeo, Michael P. Brown, Tessa Gargett, Lisa M. Ebert

**Affiliations:** 1Translational Oncology Laboratory, Centre for Cancer Biology, SA Pathology and University of South Australia, Adelaide, SA 5001, Australia; erica.yeo@mymail.unisa.edu.au (E.C.F.Y.); MichaelP.Brown@sa.gov.au (M.P.B.); Tessa.Gargett@sa.gov.au (T.G.); 2Clinical and Health Sciences, University of South Australia, Adelaide, SA 5001, Australia; 3Cancer Clinical Trials Unit, Royal Adelaide Hospital, Adelaide, SA 5000, Australia; 4Adelaide Medical School, University of Adelaide, Adelaide, SA 5000, Australia

**Keywords:** glioblastoma, cytokine, chemokine, immune suppression, microenvironment

## Abstract

Glioblastoma is the most common form of primary brain tumour in adults. For more than a decade, conventional treatment has produced a relatively modest improvement in the overall survival of glioblastoma patients. The immunosuppressive mechanisms employed by neoplastic and non-neoplastic cells within the tumour can limit treatment efficacy, and this can include the secretion of immunosuppressive cytokines and chemokines. These factors can play a significant role in immune modulation, thus disabling anti-tumour responses and contributing to tumour progression. Here, we review the complex interplay between populations of immune and tumour cells together with defined contributions by key cytokines and chemokines to these intercellular interactions. Understanding how these tumour-derived factors facilitate the crosstalk between cells may identify molecular candidates for potential immunotherapeutic targeting, which may enable better tumour control and improved patient survival.

## 1. Introduction

Glioblastoma is the most aggressive type of primary adult brain cancer, and also the most common, accounting for 80% of primary malignant brain cancers [1]. There are 10,000 and 100,000 new annual cases of glioblastoma diagnosed in the USA and worldwide, respectively [1,2]. Although glioblastoma is a rare cancer with an incidence rate of <6 per 100,000 population, it accounts for 2.5% of total cancer-related deaths, and is the first cause of cancer death among those aged between 15 and 34 years [1]. Despite its rarity, glioblastoma is a leading cause of cancer burden in Australia with 96% of the burden resulting from premature death [3]. Strikingly, the 5-year relative survival (just 4.6% at 5 years) for glioblastoma patients has remained stable over the last three decades [3,4]. Standard treatment using maximal safe resection and chemoradiotherapy confers a median survival time of 14.6 months [5]. There is no standard second-line treatment and none that extends overall survival. Recurrence is virtually inevitable. The infiltrative pattern of growth and inherent chemo-radio-resistance of glioblastoma leads to most disease recurring near the surgical margin within 6 to 9 months of treatment. Survival of recurrent glioblastoma patients is generally less than 6 months [6].

Several immunotherapeutic modalities such as peptide vaccines, dendritic cell therapy, adoptive T cell therapy, and immune checkpoint inhibitor (ICI) therapy have also been tested in glioblastoma patients. For example, initially favourable phase I and phase II clinical trial results of a peptide vaccine led to the ACT IV study. ACT IV was a randomised, double-blind, international phase III trial of the cancer vaccine, rindopepimut, and was conducted in patients with newly diagnosed EGFRvIII-mutant glioblastoma. Rindopepimut comprised an EGFRvIII-specific peptide, which was conjugated to the molecular adjuvant, keyhole limpet haemocyanin (KLH), and admixed with the cytokine, granulocyte-macrophage colony-stimulating factor (GM-CSF). The control was KLH alone. After standard treatment with debulking surgery and chemoradiotherapy, and in the absence of progression, a total of 745 eligible patients were planned to receive at least 6–12 monthly cycles of consolidation oral temozolomide chemotherapy and were randomised to receive concurrent monthly intradermal injections of rindopepimut (n = 371) or control (n = 374). However, the study was terminated for futility after a pre-planned interim analysis and, at the final analysis, no significant differences were found in the overall survival of patients according to treatment arm allocation [7].

Based on genetic evidence indicating that cytomegalovirus (CMV) sequences were detected in the majority of glioblastoma tissues, a phase 1 study was conducted using autologous CMV-specific T-cell therapy in 11 patients with recurrent glioblastoma. The therapy was safe, CMV-specific T cells were detected in resected tumour tissue from one patient, and some patients survived longer than may be expected in this poor prognosis population [8].

An intriguing small randomised phase I study of dendritic cell therapy in 12 newly diagnosed glioblastoma patients has been reported and included a set of robust pre-clinical data. In this study, autologous dendritic cells (DCs) were generated ex vivo and pulsed with cytomegalovirus phosphoprotein 65 (pp65) RNA before injection at bilateral vaccine sites of each patient. Before this immunotherapeutic intervention, study patients had been randomised to pre-conditioning of a unilateral vaccine site with the potent recall antigen tetanus/diphtheria toxoid (Td) or autologous unpulsed DCs. The investigators found that patients who were given Td had enhanced DC migration bilaterally and significantly improved survival [9].

ICI therapy has revolutionised anti-cancer therapy for many patients with solid cancers such as melanoma and kidney cancer [10,11]. However, the success of immune checkpoint inhibitor therapy depends on reinvigoration of pre-existing anti-tumour immunity, which is generally lacking in glioblastoma. Studies of tumour mutation burden suggest that only about 3% of glioblastoma patients may benefit from ICI therapy [12,13]. Indeed, the clinical results have been disappointing. For example, in the CheckMate 143 randomised phase III clinical trial, 369 patients with first recurrence of glioblastoma after standard chemoradiotherapy were randomised to receive the PD1 inhibitor, nivolumab (n = 184) or the angiogenesis inhibitor, bevacizumab (n = 185). The primary end point of overall survival was not met in this trial although the median overall survival of patients between treatment arms was comparable [14]. Even in phase I cohorts of CheckMate 143 exploring combinations of nivolumab with the CTLA4 inhibitor, ipilimumab, the response rate was low at 7% [15].

An alternative immunotherapeutic approach is chimeric antigen receptor (CAR)-T cell therapy, in which anti-tumour immunity is supplied from outside the body by genetically engineering the patients’ own peripheral blood T cells [16]. CAR-T cell therapy specific for the CD19 antigen is revolutionising the treatment of relapsed/refractory B-cell acute lymphoblastic leukaemia and lymphoma patients, and is approved in the USA, Europe and Australia for these indications. Several early phase clinical trials have demonstrated the feasibility and safety of CAR-T cell therapy for glioblastoma patients. Although the approaches tested to date provide little evidence of sustained anti-tumour activity of CAR-T cell therapy for glioblastoma [16], one striking report of a complete remission [17] illustrates two main points: (i) delivery of CAR-T cells directly to the central nervous system (CNS, via intracavitary and intraventricular administration) was feasible and safe; and (ii) targeting a single tumour antigen may allow tumour escape from immune control because this patient progressed after 8 months with tumour that had lost the tumour antigen. Hence, given the poor survival prospects of brain tumour patients after standard first-line treatment, the lack of survival-prolonging second or later line therapies, and the minimal anti-tumour activity of therapeutic immune checkpoint blockade, a strong rationale exists for CAR-T cell therapy [16]. Nevertheless, despite this early promise of CAR-T cell therapy for glioblastoma patients, the results do not mirror the remarkable success of CAR-T cell therapy for relapsed/refractory B-cell malignancies, thus indicating the clear need to improve the effectiveness of CAR-T cell therapy for solid tumours in general and brain tumours in particular. Major obstacles include limited trafficking to and infiltration of solid tumour by CAR-T cells, and a hostile tumour microenvironment (TME) that harbours immunosuppressive cells and immune checkpoint molecules, all of which can limit the anti-tumour cytotoxicity of CAR-T cells.

Accordingly, a better understanding of (i) the types of immune cells that invade into the glioblastoma tumours, (ii) the cytokines that regulate the reactivity and function of these immune cells, (iii) the chemokines that result in the attraction of these immune cells from the periphery to the tumour site, and (iv) the bi-directional crosstalk between immune cells and tumour cells, will together guide the investigation and development of new immunotherapeutic interventions, which will aim to improve the outlook of patients with these terrible diseases. This is the purpose of the current review.

## 2. The Glioblastoma TME

In common with other solid tumours, the glioblastoma microenvironment harbours an array of non-malignant (stromal) cell types in addition to the cancer cells themselves [18,19]. The main stromal cell types in glioblastoma are cells of the immune system—discussed in detail below—and cells associated with the structure and function of blood vessels (endothelial cells and pericytes) [20]. In contrast to most other tumour types, fibroblasts are not known to be a significant component of the glioblastoma TME. Vessels promote the growth and survival of glioblastoma cells, by facilitating blood perfusion and hence the provision of essential oxygen and nutrients. In addition, the perivascular zone can serve as a specialised niche to support the survival and function of glioma stem cells (GSCs), which are self-renewing, multipotent cells thought to produce the bulk of the malignant cells in glioblastoma [21]. In contrast, the role of immune cell populations is more complex, and the balance of pro-tumour versus anti-tumour populations likely plays a critical role in determining the trajectory of tumour growth and spread.

For many years, the brain was viewed as an immune privileged site, protected from the regular surveillance systems that operate in the periphery [22]. This concept was supported by a perceived lack of lymphatic vessels in the brain, thereby separating the brain from central lymphocyte circulation pathways, and the presence of the blood–brain barrier (BBB), which restricts the entry of leukocytes from the blood. However, functional lymphatic vessels have recently been discovered to line the dural sinuses of mice, and potentially analogous structures exist in human dura [23], suggesting that the brain is not in fact immunologically separate from the periphery. In addition, the BBB is frequently compromised in glioblastoma [22], and priming of tumour-specific T cells has been detected in glioblastoma patients [16]. Thus, it is clear that glioblastoma tumours interact with the immune system, but immune-mediated tumour control is likely hampered by an overwhelmingly immunosuppressive TME.

Cells of the myeloid lineage are a major component of the glioblastoma TME [20,24,25,26]. In fact, these cells are reported to constitute a remarkable 30–50% of the glioblastoma tumour mass. Myeloid cell types within the glioblastoma TME include brain-resident microglia and infiltrating macrophages, which are collectively referred to as glioma-associated microglia and macrophages (GAMs), as well as myeloid-derived suppressor cells (MDSCs). Microglia are derived from primitive yolk sac progenitors that enter the brain during embryogenesis and reside as a local resident population throughout life [25,26]. They play many critical roles under conditions of homeostasis, including synaptic pruning and the regulation of sleep and memory, as well as serving as local sensors of neuronal damage and infection. In contrast, infiltrating macrophages are thought to enter the tumour as blood-borne monocytes, which are recruited in response to inflammatory stimuli, and then differentiate to macrophages once they enter the TME [19,25]. Finally, MDSCs are immature myeloid lineage cells with inherent immunosuppressive properties [19,27]. They arise through a pathological (tumour-driven) block in normal myeloid differentiation pathways, leading to the accumulation of an abnormal population of partially differentiated myeloid cells. MDSCs exploit a number of immunosuppressive mechanisms to inhibit adaptive immune responses, including the depletion of nutrients required for effective T cell responses, the generation of oxidative stress conditions that inhibit T cell function, and the activation and expansion of regulatory T cells (Tregs) [19,27].

In contrast to the inherent immunosuppressive properties of MDSCs, macrophages and microglia are more plastic cell types that can be readily polarised according to their local environment, resulting in highly divergent functions [19,25,27]. According to the M1/M2 paradigm, ‘classically activated’ macrophages (M1) assume an inflammatory phenotype characterised by efficient phagocytosis and antigen presentation, and abundant production of pro-inflammatory cytokines. In contrast, ‘alternatively activated’ macrophages (M2) largely produce anti-inflammatory cytokines and support tissue remodelling and matrix deposition. However, this proposed dichotomy is largely based on in vitro studies, and it is likely that macrophages in tissues rarely exist in such clearly defined states. Indeed, although GAMs clearly can express markers of the immunosuppressive M2 phenotype, including TGF-β, IL-10, CD163 and CD204, unbiased transcriptomic analyses characterised GAMs in patient tissues as more in keeping with a non-polarised M0 phenotype [28], or a mixed M1/M2 phenotype [29].

GAMs are most commonly considered to have a pro-tumorigenic function. For example, they possess multiple immunosuppressive properties, secrete factors that actively promote tumour cell proliferation and invasion, and in certain animal models it has been demonstrated that depletion of GAMs can significantly reduce tumour growth [19,25,26,30,31]. Furthermore, in patient glioblastoma tissues, the proportion of M2 macrophages is reported to positively correlate with the rate of tumour cell proliferation [32]. However, several other studies have shown conflicting results. For example, in some animal models GAM depletion actually enhances tumour growth [25,33], while a high frequency of either total or M2-phenotype GAMs in patient glioblastoma tissues correlate with improved survival [29]. The role of GAMs in the growth and progression of glioblastomas is therefore likely to be complex and highly context-dependent and requires further study.

Although GAMs represent the predominant immune cell population in glioblastoma, significant populations of lymphocytes are also present. These are primarily T cells, although natural killer (NK) cells and B cells have also been identified in human glioblastomas, the latter being relatively rare [19]. The T cell population in glioblastoma generally displays a profoundly exhausted phenotype, characterised by expression of LAG3, TIGIT, CD39 and especially programmed cell death 1 (PD1) [34]. T cell anti-tumour activity can also be inhibited by indoleamine 2,3-dioxygenase (IDO), an enzyme present in the TME responsible for catalysing the oxidation of tryptophan to downstream metabolites belonging to the kynurenine pathway. This can, through a variety of mechanisms, lead to T cell dysfunction, an effect that is particularly pronounced in the setting of advanced age [35,36]. Furthermore, Tregs are enriched in glioblastoma lesions compared to peripheral blood, and are expected to further inhibit the function of effector T cells, as well as NK cells [19,37]. This severely immunosuppressed microenvironment likely contributes to the apparent inability of infiltrating T cells to control tumour growth. This effect is compounded by the inherent low immunogenicity of glioblastoma tumours, which generally lack the high mutation rate thought to be required for robust anti-tumour T cell responses [38]. However, it is worth noting that a high effector CD8+ T cell frequency in patient glioblastoma tissues is associated with prolonged survival [39]. In addition, ICI therapies, which promote anti-tumour T cell responses, can induce regression of glioblastomas harbouring germline mismatch repair deficiency, which are characterised by a greatly elevated mutation rate [12]. Thus T cells may have the potential to control glioblastoma growth in circumstances where their frequency and function are optimal, highlighting the therapeutic potential of T cell-based therapies in this disease.

## 3. Cytokines and Chemokines That Regulate the Immune Microenvironment of Glioblastoma

### 3.1. Introduction

Intercellular communication between immune cell populations is critical for orchestration of the immune system. This communication can be facilitated by cell-to-cell contact or soluble mediators via cytokines and chemokines. Cytokines are signalling proteins (<70 kDa) secreted by cells to regulate embryonic development, haematopoiesis and immune responses through local and systemic communication [40,41,42]. Cytokines can act on the cells that secrete them (autocrine activity); on cells in close proximity (paracrine activity); or on cells at a distant site (endocrine activity). Chemokines (8–14 kDa) belong to a family of chemotactic cytokines, which were initially reported for their ability to facilitate leucocyte migration [40,43]. However, they also play additional roles in influencing T cell differentiation, activating integrins for leucocyte extravasation and tumour progression [40,44]. The chemokine family can be divided into four groups based on the relative position of the two cysteine residues in the N-terminal domain: C chemokines with only one cysteine, CC chemokines with two adjacent cysteines, CXC chemokines with one amino acid between the two cysteines, and CX3C chemokines with three amino acids between the two cysteines [45]. This structural distinction is the basis of the systematic nomenclature used for chemokines and their receptors.

The binding of a cytokine or chemokine to its cognate receptor on the cell surface can trigger multiple signalling cascades that result in immediate effector functions and the expression of genes necessary for altered cell function. Cytokines bind to receptors that have intrinsic kinase activity or are associated with kinases, whereas chemokines are recognised by G-protein coupled receptors [40,46]. Chemokine receptors are classified similarly to their respective ligands, such as XCRs, CCRs, CXCRs and CX3CRs. However, certain chemokines do not bind to the classical chemokine receptors, but instead to a different subset of receptors described as atypical chemokine receptors (ACKRs) [45,47]. These receptors, which are not directly coupled to G proteins, do not trigger the classical chemokine-induced signalling pathways but are involved in the regulation of chemokine availability and conventional chemokine receptor-mediated signalling. Further adding to the complexity of the chemokine system are the phenomena of redundancy and pleiotropy, which may allow efficient immune responses but can complicate our understanding of the biological roles of each chemokine or chemokine receptor. Chemokine redundancy refers to the fact that many different chemokines can induce similar effects through the same receptor whereas pleiotropy refers to the ability of an individual chemokine to exert many different types of effects through different receptors expressed on a variety of cells [48].

Cytokines and chemokines can be upregulated in a variety of human malignancies, but their roles are multifaceted with some exhibiting anti-tumour effects in certain cancers and contrasting effects in other cancers. As these soluble mediators exhibit immunomodulatory effects and influence the microenvironmental landscape of glioblastoma, we aim to dissect the specific roles of key cytokines and chemokines in glioblastoma as elucidated by recent studies (Figure 1, Table 1).

### 3.2. CCL2

CCL2 was first identified from the supernatant of glioma cells and originally referred to as monocyte chemoattractant protein-1 (MCP-1) based on its chemotactic activity for monocytes [49]. It is now known to be expressed by an array of cells, such as endothelial cells, epithelial cells, smooth muscle cells, fibroblasts, astrocytes, neurons, myeloid cells and T cells, under physiological conditions as well as pathological states associated with inflammatory and neurodegenerative conditions [50,51,52,53,54,55,56,57,58]. Inflammatory stimuli that induce CCL2 expression include interferon-gamma (IFN-γ), interleukin-1, -4, -6 (IL-1, IL-4, IL-6), transforming growth factor-beta (TGF-β) and tumour necrosis factor-alpha (TNF-α) [56,59,60,61]. CCL2 attracts monocytes and other immune cell populations by signalling through CCR2 and CCR4 [62,63,64,65]. Apart from CCL2, CCR2 additionally recognises CCL7, CCL8, CCL13 and CCL16, while CCR4 also recognises CCL17 and CCL22 [66].

CCL2 expression has been detected in a plethora of cancer types including glioblastoma. CCL2 expression is significantly higher in glioblastoma patient tumour samples and cell lines compared to healthy brain tissue, and its gene expression has a negative correlation with overall survival of patients [67]. Within the glioblastoma TME, GAMs and tumour cells have been identified as the main sources of CCL2, though evidence suggests GAM-secreted CCL2 has a more significant impact, possibly due to higher levels being produced [67,68]. Factors such as hypoxia and necrotic cells can enhance the levels of CCL2 within the TME [68,69], whereas treatment with chemotherapeutic agents temozolomide (TMZ) and 1,3-bis(2-chloroethyl)-1-nitrosourea (BCNU) can reduce CCL2 production [70].

The CCL2/CCR2/CCR4 pathway exerts immunomodulatory effects on glioblastoma through the recruitment of CCR2+ GAMs, CCR2+ MDSCs and CCR4+ Tregs, and these immunosuppressive cells can contribute to immune escape by attenuating the effector T cell response. The use of CCL2-neutralising antibodies (anti-CCL2) can prevent the recruitment of GAMs and MDSCs and prolong animal survival [68,71], with the addition of TMZ further improving overall survival compared to monotherapies alone [68]. Although no difference in Treg influx was observed with anti-CCL2 [68], suggesting possible compensatory roles of other chemokines in Treg mobilisation, *Ccl2*-/- mice intracranially implanted with glioblastoma tumour cells had reduced infiltrating Tregs and monocytic MDSCs [67]. Interestingly, these tumour cells secrete CCL2, yet failed to induce maximal immune cell recruitment to the tumour site, implying that TME-derived CCL2 may play a greater role in trafficking of these immunosuppressive cells [67,68]. *Ccr2*-deficiency can also reduce the influx of monocytic MDSCs [67] and GAMs [72], though a GAM population could still be identified, likely due to recruitment independently of the CCL2/CCR2 signalling pathway. In mice bearing anti-PD1-resistant glioblastoma, the CCR2 antagonist CCX872 enhanced the efficacy of anti-PD1 via a reduction in MDSCs and a concomitant increase in functional T cells within the tumours, significantly improving the overall survival of mice [73]. Finally, targeting CCR4 through gene knockout or use of the antagonist C 021, reduced the recruitment of Tregs to the brain [67]. Collectively, these studies highlight a key role for the CCL2 ligand and both of its receptors (CCR2 and CCR4) in promoting the recruitment of immunosuppressive myeloid and Treg populations to glioblastoma.

### 3.3. CCL5

CCL5, previously known as Regulated upon Activation, Normal T cell Expressed and Secreted (RANTES), is found to be expressed by epithelial cells, platelets, macrophages, T cells and NK cells [74,75,76,77]. CCL5 plays a role in the recruitment of immune cells including monocytes, macrophages, eosinophils, basophils, dendritic cells and T cells to inflammatory sites [74,78,79,80], and also the retention of tissue-resident lymphocytes within non-lymphoid tissues [77]. CCL5 primarily acts through CCR5, though it is also recognised by CCR1, CCR3, CCR4 and CD44 [81,82]. CCR5, which is also a receptor for CCL3, CCL4 and CCL8 [66,83], is well known for its role as a co-receptor utilised by human immunodeficiency virus (HIV) to enter into CD4+ T cells [84].

CCL5 has been identified to be overexpressed in glioblastoma. Patient tumours showed elevated *CCL5* expression compared to the normal brain [85], suggesting the expression is upregulated in the course of malignant transformation. CCL5 can act as a prognostic measure of glioblastoma patient survival, as *CCL5* overexpression is associated with shorter overall survival [86]. Within the glioblastoma TME, CCL5 produced by GAMs, mesenchymal stem cells (MSCs) and tumour cells signals through CCR1, CCR5 and CD44 [86,87,88,89]. Similar to its ligand, CCR5 expression in glioblastoma has also been associated with poor survival [90], and this expression can be upregulated on tumour cells under hypoxic conditions as an adaptive mechanism for the adverse environment [91].

CCL5 and its receptors CCR1 and CCR5 are key players that induce microglia migration to the tumour site and subsequent changes in phenotype [92]. GSCs isolated from patient glioblastoma tumours secrete CCL5 into the supernatant and induce the migration of primary human microglia [89]. This effect can be reduced, albeit modestly, with the addition of anti-CCL5. Apart from mediating microglia migration, CCR5 also upregulates arginase-1 (*ARG1*) and interleukin-10 (*IL10*) gene expression, which are well-reported markers associated with the immunosuppressive M2 phenotype [92]. Maraviroc, a CCR5 blocker approved by the United States Food and Drug Administration (FDA) for the treatment of HIV infection, can attenuate glioma-induced microglial migration and reverse M2 polarisation [92]. Interestingly, an orthotopic syngeneic murine glioma model lacking either *Ccr1* or *Ccr5* did not reduce the number of infiltrating GAMs, which suggested redundancy in the mechanisms of recruitment of these cells [88]. Indeed, it was found that primary cultures of microglia co-expressed CCR1 and CCR5, and hence either receptor is capable of trafficking GAMs to the tumour. This was further demonstrated with Met-CCL5, a dual CCR1 and CCR5 antagonist, which can significantly reduce the migration of *Ccr1*−/− or *Ccr5*−/− microglia in vitro. However, due to the lack of an appropriate model system with combined deficiency of both receptors, further investigation is still required to validate this hypothesis.

Apart from its immunomodulatory effects, CCL5 also directly affects glioblastoma tumour cells. With the expression of both CCL5 and its receptor CCR5, tumour cells can use the autocrine signalling circuit to regulate their own proliferation, through activation of the phosphoinositide 3-kinase (PI3K)/AKT pathway. Targeting this signalling pathway with pharmacological inhibitors such as CCR5 blocker Maraviroc, PI3K/AKT inhibitor LY294002 and PI3K inhibitor NVP-BKM120 can suppress tumour cell proliferation [87,90,92,93]. The link between the CCL5/CCR5 pathway and tumour growth is also evident in vivo, as knockdown of *CCR5* in tumour cells can significantly reduce tumour size and expression of the proliferation marker Ki67 [90]. The CCL5/CCR5 pathway also supports tumour cell invasion by activating downstream pathways such as PI3K/AKT and calcium/calmodulin-dependent protein kinase II (CaMKII) that ultimately result in the production of matrix metalloproteinase-2 (MMP-2) and -9 (MMP-9) [86,90,91]. These enzymes degrade extracellular matrix (ECM) barriers, enabling an active migratory process with the penetration of tumour cells deeper into surrounding brain tissue [94]. By inhibiting CCR5 or its downstream migratory pathways, the production of MMPs was reduced, limiting the invasion capacity of tumour cells [86,90,91]. There are also reports on the role of CCL5 in activating the mammalian target of rapamycin (mTOR) pathway, which has been shown to be critical for the maintenance of GSCs and the survival of mesenchymal glioblastoma tumour cells [85,95,96]. To elucidate the role of mTOR signalling in GSCs, mTOR inhibitors AZD2014 or PP242 were added to GSCs, resulting in attenuation of self-renewal, sphere forming ability and radioresistance [95,96], all of which may represent cancer stem cell properties.

### 3.4. CXCL12

CXCL12, also known as pre-B cell growth factor (PBGF) and stromal cell-derived factor-1 (SDF-1), is involved in processes such as embryogenesis, lymphopoiesis, wound healing and T cell homing [97,98,99]. CXCL12 is produced by osteoblasts, fibroblasts, dendritic cells, monocytes, glial cells and neuronal cells [100,101]. Factors such as hypoxia and growth inhibition can upregulate CXCL12 expression [102]. CXCL12 is recognised by CXCR4 and ACKR3 found on haematopoietic cells, neuronal cells, endothelial cells and epithelial cells [101,103,104,105]. CXCL12 is the only chemokine ligand for CXCR4, and this restricted receptor selectivity is unique among the promiscuous chemokine-receptor relationships [106]. ACKR3, initially known as RDC-1 and CXCR7, is classified under the atypical chemokine receptor family due to its β-arrestin-dependent pathways, making it distinct from conventional chemokine receptor family members that use G-protein-dependent pathways [107]. ACKR3, which also binds to CXCL11, maintains CXCL11 and CXCL12 gradients in the environment by ligand sequestration [108]. Similar to CCR5, both CXCR4 and ACKR3 can also serve as co-receptors for HIV [109,110].

CXCL12 has been implicated in the progression of glioblastoma, with the expression of CXCL12 rarely identified in low-grade gliomas. Glioblastoma tumour cells are the main contributors of CXCL12 within the TME, with hypoxic stimuli, TMZ and irradiation further exacerbating production [111,112,113]. CXCR4 and ACKR3 in glioblastoma are found on tumour cells, GAMs and endothelial cells [114,115,116]. Although CXCR4 and ACKR3 can be co-expressed on tumour cells, CXCR4 tends to be more highly expressed on GSCs, whereas ACKR3 is detected at higher levels on differentiated tumour cells [115,117,118,119,120,121]. Upon differentiation in vitro, GSCs downregulate expression of CXCR4 and stem-cell markers and increase expression of ACKR3 and differentiated astroglial marker glial fibrillary acidic protein (GFAP) [115], further supporting the influence of tumour cell differentiation on the expression of CXCL12 receptors. Despite ACKR3 being recognised as a CXCL12 scavenging receptor, the functional role of ACKR3 in glioblastoma is complex because it might regulate CXCL12 levels in the TME and concomitantly modulate CXCR4 signalling.

CXCL12 promotes the influx of bone marrow-derived cells (BMDCs) into glioblastoma tumours under hypoxic conditions to initiate neovascularisation. Tumour hypoxia is a key feature in glioblastoma, and this can be further exacerbated by vascular damage and reduced perfusion caused by irradiation [114]. Transcription factors known as hypoxia-inducible factors (HIFs) become stabilised in response to hypoxic stress, allowing tumour cells to initiate adaptive changes such as new blood vessel formation to survive in the adverse microenvironment. This occurs through increased production of pro-angiogenic factors such as vascular endothelial growth factor (VEGF) and CXCL12 [122,123]. CXCL12 recruits CXCR4+ BMDCs to the tumour, including vascular progenitor cells that can differentiate and incorporate into new blood vessels, but also MMP-9-expressing myeloid cells that indirectly regulate tumour angiogenesis [114,124]. MMP-9 is critical for the induction of angiogenesis because it degrades the ECM and releases bound VEGF, thus enabling interaction with its receptor on endothelial cells [112].

Glioblastoma virtually always recurs following standard treatment including radiotherapy. Recurrences are often observed within the radiation field [114], and radiotherapy itself can help to re-establish a functional tumour vasculature. To elucidate the importance of BMDC-dependent neovascularisation following hypoxia or irradiation, HIF-1 activity was reduced by gene silencing or the pharmacological inhibitor NSC-134754 [114]. Investigators found that the tumours displayed reduced infiltration with CD45+ myeloid cells, diminished angiogenic activity, and persistent shrinkage with no regrowth, which may result from insufficient MMP-9 to induce neovascularization. The diminished angiogenic response was also seen in tumours established from *Mmp-9* knockout cells or tumours treated with CXCR4 inhibitor AMD3100 [112]. These data revealed that tumour growth following exposure to hypoxia is dependent on the recruitment of CD45+ CXCR4+ MMP-9-expressing myeloid cells to restore the tumour vasculature. Apart from being a chemoattractant, CXCL12 could also polarise the recruited CXCR4+ myeloid cells into the M2 immunosuppressive phenotype. This is supported by observations seen in an orthotopic human U87 xenograft model treated with CXCR4 antagonist peptide R, where infiltrating myeloid cells displayed increased M1 marker expression compared to untreated mice [125].

The binding of CXCL12 to CXCR4 or ACKR3 on tumour cells has contrasting effects. Numerous studies have demonstrated a role for CXCR4 in mediating tumour cell proliferation, through the use of CXCR4 antagonists AMD3100, peptide R and PRX177561, or by downregulating CXCR4 expression [114,117,120,121,125,126,127]. This proliferative response is likely to be mediated by ERK1/2 and AKT pathways [126]. CXCL12 can also induce the directional migration of tumour cells [120,121,125]. It was previously mentioned that CXCR4 expression on tumour cells is lost upon cell differentiation, thus proposing a role of CXCR4 in sustaining a stem cell phenotype. Indeed, CXCR4 was found to be critical for the survival and self-renewal of GSCs, with CXCR4 blockade leading to increased apoptosis and differentiation [118,120,121]. Investigators have also discovered a role for CXCL12 in mobilising GSCs into protective niches that make them resistant to therapy, although it is not clear which CXCL12 receptor is responsible for this effect [119]. The maintenance of tumour vasculature appears to involve CXCR4 activity, with CXCR4 antagonists reducing GSC production of VEGF, tumour cells’ ability to mimic vascular structures and the number of CD31+ or VEGF+ cells intratumourally [120,121,125]. On the contrary, no role could be identified for ACKR3 in promoting proliferation, survival or migration of tumour cells [115,118]. Instead, this receptor appears to mediate resistance to drug-induced apoptosis. Thus, CXCL12 prevents tumour cell apoptosis following camptothecin or TMZ treatment, but drug sensitivity can be re-established in the presence of the ACKR3 antagonist CCX733 [111,115].

### 3.5. Interleukin-6 (IL-6)

IL-6 is a pleiotropic cytokine and was first described by such terms as B-cell stimulatory factor-2 (BSF-2), interferon-β2 (IFN- β2), hybridoma growth factor (HGF) and hepatocyte-stimulating factor (HSF), based on its many roles [128]. It is produced by many cell types such as fibroblasts, endothelial cells, monocytes, neutrophils, T cells and B cells [129,130,131,132,133,134]. IL-6 regulates various biological functions including the acute phase response, defence against infections, leucocyte infiltration at sites of inflammation, leucocyte maturation, and endothelial cell properties [135,136,137,138,139,140,141,142]. The IL-6 signalling pathway begins with IL-6 binding to membrane-bound IL-6R (also known as gp80) found on hepatocytes, neutrophils, monocytes, B cells and T cells [143,144,145]. The IL-6-IL-6R complex thereafter binds to membrane-bound gp130, shown to be expressed ubiquitously by all cells [146], to form an activated IL-6 receptor that initiates intracellular signalling. These events form the classic IL-6 signalling pathway, which induces anti-inflammatory responses [147,148]. However, IL-6 can also act via the trans-signalling pathway, with IL-6 binding to soluble forms of IL-6R and the complex binding to membrane-bound gp130 [146]. This allows IL-6 to activate target cells lacking the membrane-bound IL-6R and subsequently induces pro-inflammatory responses [148,149]. Both classic and trans-signalling pathways lead to the activation of Janus kinases (JAKs) and signal transducer and activator of transcription (STATs) that modulate cellular responses [146].

IL-6 may represent a prognostic factor in glioblastoma patients because high *IL-6* gene expression was associated with poor survival according to the datasets derived from The Cancer Genome Atlas (TCGA) and the Repository of Molecular Brain Neoplasia Data (REMBRANDT) [150]. This association was further supported by IL-6 gene and protein expression analyses conducted by other research groups [151,152,153]. IL-6 within the glioblastoma TME is secreted by tumour cells, GAMs and tumour-associated endothelial cells [152,154], and its expression can be enhanced by hypoxia, chemotherapy and radiotherapy [155,156,157]. IL-6 mediates its activity through IL-6R expressed on tumour cells and GAMs [150,158].

IL-6 promotes immunosuppressive GAMs and suppresses T cell functions. Exposure of GAMs to IL-6 results in an M2 phenotype with the expression of immunosuppressive molecules PD-L1, B7-H4 and Arg-1, and lineage markers such as CD163 and CD206 [150,153]. PD-L1 and B7-H4, both belonging to the B7 family of immune checkpoint molecules, have been implicated in mediating T cell dysfunction by suppressing T cell proliferation and survival [159,160]. Treatment with anti-IL-6 siltuximab, anti-IL-6R (gp80) tocilizumab or STAT3 inhibitor stattic could abrogate PD-L1 expression and T cell apoptosis [153], demonstrating the role of IL-6 signalling in regulating T cell responses in glioblastoma through GAMs. Interestingly, investigators found the therapeutic benefit of anti-IL-6 antibody to be dependent on CD8+ T cells, as CD8-depleted mice did not demonstrate the reduction in tumour size or increase in survival that were observed in control mice and CD4-depleted mice following anti-IL-6 treatment. The expression of Arg-1 on GAMs can also suppress T cell responses [161]. Arg-1 can be induced by IL-6 synergistically with colony stimulating factor-1 (CSF-1) through the activation of AKT/mTOR pathway [150].

IL-6 also sustains tumour progression by acting directly on glioblastoma cells to induce anti-apoptotic pathways and promote invasion. Autophagy serves as an adaptive response to cellular stresses such as nutrient deprivation, allowing damaged cellular material to be engulfed and lysed, leading to a turnover of components that can help to sustain cellular metabolism [162]. Autophagy initially acts as a tumour suppressor in healthy cells, but conversely acts as a tumour promoter once cancer is established, allowing tumour cells to meet their increasing energy demands required for proliferation and to survive under hypoxic conditions [163]. The contribution of autophagy to tumour progression was evident in grade III gliomas, in which IL-6, HIF1A (marker of hypoxia) and LC3B (marker of autophagy) were found to co-localise in hypoxic regions within the tumours [157]. This observation suggests that hypoxic glioblastoma cells rely on IL-6 to initiate autophagy to support growth in a nutrient-deprived milieu. Indeed, the application of anti-IL-6 and anti-IL-6R tocilizumab reduced hypoxia- or TMZ-induced autophagy and caused significant tumour cell apoptosis [157]. IL-6 can also initiate anti-apoptotic pathways following irradiation. Although irradiation may be effective in eliminating some tumour cells, IL-6-expressing tumour cells were unaffected [164], and moreover, enhanced the acquisition of a radioresistant phenotype by repressing irradiation-induced DNA damage [165]. IL-6 also has additional roles in mediating tumour cell invasion by upregulating the expression of MMPs and fascin-1 [166,167]. As mentioned previously, MMPs are involved in degrading the ECM barriers. On the contrary, fascin-1 is an actin-bundling protein that regulates actin cytoskeleton remodelling, and thus it is crucial for cellular protrusions that drive an invasive phenotype.

### 3.6. Transforming Growth Factor-Beta (TGF-β)

TGF-β signalling is involved in the regulation of cell growth and differentiation, and this function is lost upon tumorigenesis [168]. The TGF-β family consists of three TGF-β isoforms: TGF-β1, TGF-β2 and TGF-β3. These isoforms are synthesised as precursor molecules bound to latency associated proteins (LAPs) and latent TGF-β binding protein (LTBP) and are released from the complex by enzymatic cleavage to form mature TGF-β. TGF-β is expressed in all tissues by many cell types including leucocytes [169]. The TGF-β signalling cascade begins with the binding of TGF-β to TGF-β type II serine/threonine kinase receptor (TβRII), and recruitment of TGF-β type I receptor (TβRI) to the complex [168]. The activated complex then initiates Smad and non-Smad pathways to modulate TGF-β target genes. The canonical TGF-β signalling pathway activates Smad proteins as signal transducers, whereas the alternative pathway activates PI3K/AKT pathways, mitogen-activated protein kinase (MAPK) pathways and Rho family GTPases.

TGF-β is expressed at low levels in the brain, but it is greatly enhanced in glioblastoma with TGF-β1 and TGF-β2 being the predominant isoforms [170,171]. Upon further analysis of glioblastoma molecular subtypes, higher TGF-β1 was observed in the mesenchymal subtype and higher TGF-β2 and TGF-β3 in classical and mesenchymal subtypes. TGF-β1 expression, but not TGF-β2 and TGF-β3, displayed a significant negative correlation with patient survival [171]. Tumour cells, endothelial cells, GAMS and Tregs secrete TGF-β, which through autocrine or paracrine mechanisms can contribute to tumour growth, immunosuppression and angiogenesis [172,173].

TGF-β causes defects in endogenous anti-tumour immunity by suppressing effector cell functions. Inhibition of TβRI by SD-208 in vivo resulted in greater NK cell and CD8+ T cell infiltration into the tumour, suggesting an inhibitory effect of TGF-β on effector cell recruitment [174]. In addition, cumulative studies have identified a role for TGF-β in regulating the expression of NKG2D and its ligands in glioblastoma to facilitate immune escape. The expression of NKG2D ligands under physiological conditions is generally low but can be induced upon cellular stresses such as infection and malignant transformation [175]. NK cells and CD8+ T cells expressing NKG2D can recognise glioblastoma tumour cells by binding to NKG2D ligands, triggering a cytolytic immune response. However, TGF-β downregulates the expression of NKG2D and its ligands, thereby allowing tumour cells to evade immune surveillance [176]. TGF-β also suppresses CD8+ T cell anti-tumour cytolytic activity. CD107a, also known as lysosomal-associated membrane protein-1 (LAMP-1), is a functional marker of cytotoxic NK cell and CD8+ T cell degranulation, with expression found on lysosomal membranes that encapsulate cytotoxic granules [177,178]. Upon recognition of a target cell, secretory lysosomes are trafficked to the surface to release cytotoxic granules, with surface CD107a expression upregulated as lysosomes fuse with the cell membrane. However, the expression of surface CD107a on CD8+ T cells co-cultured with tumour cells was decreased in the presence of TGF-β, implicating TGF-β in regulating the release of cytotoxic granules [179]. Additionally, TGF-β repressed the release of pro-inflammatory cytokines IFN-γ and TNF-α by lymphocytes, which are crucial mediators of anti-tumour effects, thus leading to a restricted response against grade III glioma cells [174]. As expected, TβRI inhibitors SX-007 and SD-208 can restore effector cell responses to tumour cells with greater release of cytotoxic granules and pro-inflammatory cytokines, delineating the negative regulation imposed by TGF-β on anti-tumour effector cell activity [174,179].

The TGF-β pathway also directly regulates the malignant potential of glioblastoma tumour cells. The proliferative response mediated by TGF-β can differ between each glioblastoma cell line, with some gaining or losing proliferative functions, while others had no impact [180,181,182,183]. However, it is important to mention that TGF-β inhibitor SB-431542 and TβRI inhibitor LY2109761 blocked the effect of TGF-β in proliferative cell lines, thus implicating TGF-β in tumour cell proliferation [180,182]. TGF-β contributes to the migratory potential of tumour cells by reducing adhesive properties and mediating cellular morphological changes to promote tumour cell motility [182,184,185]. TGF-β also induces invasive capability through MMP expression and activity [185,186]. Characteristics of GSCs, such as self-renewal, sphere formation, tumour-initiating capacity and radioresistance, can be promoted by TGF-β and suppressed when the signalling pathway is inhibited [187,188,189,190]. Moreover, TGF-β also indirectly regulates tumour angiogenesis through VEGF expression [182].

### 3.7. Colony Stimulating Factor-1 (CSF-1)

CSF-1, also known as macrophage colony-stimulating factor (M-CSF), is a growth factor that regulates the survival, proliferation and differentiation of mononuclear phagocytic cells such as monocytes and macrophages [191]. CSF-1 is also implicated in pathological conditions including bone disease, inflammatory disease and cancer. CSF-1 is produced by osteoblasts, fibroblasts and endothelial cells, either as a soluble form to mediate humoral regulation or transmembrane form for local actions [192,193,194]. CSF-1 acts by binding to colony stimulating factor-1 receptor (CSF-1R) expressed on monocytes, macrophages, and dendritic cells [191]. However, CSF-1R also recognises a second ligand known as IL-34 [195].

*CSF1* expression is significantly upregulated in glioblastoma compared to the normal brain [196]. Tumour cells are identified as the source of CSF-1 in glioblastoma, and production can be enhanced following irradiation [197]. *IL34*, however, is downregulated in glioblastoma and thus does not play a significant role in CSF-1R signalling in glioblastoma [196]. As CSF-1 receptor is only expressed on GAMs, tumour cells can be presumed to secrete CSF-1 to regulate GAM behaviour and thus influence tumour biology [198].

Targeting CSF-1 signalling can reverse the pro-tumour phenotype of GAMs. CSF-1 promotes the proliferation and survival of GAMs, with CSF-1R inhibitors BLZ945 and PLX3397 inducing anti-proliferative and cytotoxic effects in vitro [197,199]. In vivo, however, CSF-1R inhibitor only depleted microglia in the adjacent normal brain but not GAMs within proneural glioblastoma xenografts and murine gliomas. It was identified in culture that tumour-derived factors such as IFN-γ and GM-CSF can protect GAMs from CSF-1R blockade [199,200]. Nevertheless, further analysis of these surviving GAMs revealed a decrease in M2-associated genes and enhanced phagocytic abilities, thus blunting their tumour-promoting functions. This decrease in M2 polarisation also repressed the pro-angiogenic function of GAMs, thus inhibiting the re-establishment of tumour vasculature after radiation [197,201]. Finally, T cell influx was augmented following re-education of GAMs in the context of CSF-1R blockade, likely through GAM-secreted cytokines and chemokines that promote an environment favourable for T cell activity [202].

The crosstalk between GAMs and tumour cells via CSF-1 can indirectly regulate glioma development. Blockade of CSF-1R reduced tumour cell proliferation and enhanced apoptosis in vivo, despite minimal cytotoxic effects on tumour cells in vitro, highlighting the importance of TME-derived GAMs in driving tumour cell progression [199,200]. Furthermore, CSF-1 also regulates glioblastoma aggressiveness, with histological analysis of CSF-1R inhibitor-treated tumours revealing grade II and III glioma features, as opposed to the grade IV glioblastoma features seen in vehicle-treated mice [200]. An autochthonous glioma model, with characteristics of the proneural subtype of human high-grade gliomas, also emphasized the importance of CSF-1 signalling in de novo gliomagenesis, with CSF-1 overexpression significantly increasing the formation of high-grade gliomas [196]. This crosstalk between glioblastoma tumour cells and GAMs can also regulate tumour cell invasiveness. Once recruited to the tumour site and activated via CSF-1R, pro-tumourigenic GAMs secrete epidermal growth factor (EGF) to stimulate tumour cell invasion [203]. Consequently, targeting CSF-1R produced less invasive tumours with clearly delineated borders, thus illustrating a synergistic interaction between tumour cells and GAMs [199,203].

### 3.8. Additional Soluble Factors with Immunomodulatory Roles in Glioblastoma

Thus far, this review has focussed on cytokines and chemokines whose role in glioblastoma has been studied in some detail. Several other soluble factors have been less extensively studied, but evidence still exists for an immunomodulatory role in the context of glioblastoma. A few examples are mentioned below.

IL-10 plays an important role in limiting excessive inflammatory responses that can contribute to tissue damage. IL-10 represses the immune system by downregulating antigen presentation, blocking the maturation of dendritic cells and sustaining the Treg population, all of which can hinder T cell effector function [204,205]. *IL-10* mRNA expression is found to be elevated in glioblastoma, with M2-like GAMs and GSCs reported to contribute to its production [206,207,208]. Within glioblastoma tumours, exposure to IL-10 upregulates PD-L1 expression on GAMs, with the functional consequence of inducing T cell apoptosis [209]. However, targeting IL-10 or IL-10 receptor can diminish PD-L1 expression. IL-10 is also directly implicated in glioblastoma cell growth and proliferation [210].

Macrophage migration inhibitory factor (MIF) is another key soluble factor driving immune suppression. The levels of MIF were found to be significantly elevated in gliomas compared to the normal brain, with levels correlating with glioma grade [211]. MIF is produced by glioblastoma tumour cells including GSCs [211,212], and levels are significantly enhanced under hypoxia [213]. MIF exerts its effects through interaction with CD74, CXCR2 and CXCR4 found on MDSCs and tumour cells [213,214]. MIF signalling through monocytic MDSCs was found to induce CCL2 secretion [214], which could drive the recruitment of immunosuppressive immune cells as previously discussed. It was also found that MIF can elevate Arg-1 expression in MDSCs through CXCR2 [212]. These MDSC functions can be reduced upon interference of MIF signalling, with effector T cell responses significantly enhanced in tumour-bearing mice with attenuated MIF [212,214]. Just like TGF-β, MIF has also been identified to attenuate the activity of NK cells and CD8+ T cells by downregulating the expression of NKG2D [211]. Furthermore, MIF has been implicated in the progression of vascular mimicry, in which tumour cells organise in vascular-like structures to receive sufficient blood supply [213].

Originally isolated from the bone but later found to be widely distributed, osteopontin (OPN) contributes to bone remodelling and the migration of immune cells [215]. OPN has been linked to the progression of glioblastoma, as increased gene expression was reported in glioblastoma tumours relative to healthy brain tissues and was associated with poorer prognosis [216]. Produced by glioblastoma tumour cells and GAMs, OPN engages with integrin proteins such as α_V_β_5_ and CD44 [216,217]. OPN drives the recruitment of GAMs to the TME and supports an immunosuppressive M2 phenotype [216,218]. Both tumour- and stromal-derived OPN play a significant role in mediating immune suppression, as OPN deficiency either in the tumour or stromal compartment reduced the number of infiltrating GAMs and enhanced T cell effector activity [216].

## 4. Concluding Remarks and Future Perspectives

Patients diagnosed with glioblastoma face a poor prognosis and limited treatment options and they are yet to benefit from the immunotherapy revolution that is transforming the treatment of other cancer types. A deeper understanding of the tumour/immune microenvironment is required to underpin investigations of new therapies for glioblastoma patients.

T cells so far represent the immune population with the most demonstrated potential for treating cancer; either in the form of endogenous T cells empowered by molecules such as immune checkpoint inhibitors, or exogenous T cells enhanced with tumour specificity via chimeric antigen receptor technology. Yet, neither of these approaches, which have proved so successful in the treatment of metastatic melanoma and acute lymphoblastic leukemia, respectively, have demonstrated efficacy for glioblastoma. Immune checkpoint blockade benefits a small minority of the glioblastoma patient population [12], and CAR-T cell clinical trials are still in their infancy, with only four phase 1 trials and limited clinical responses reported to date [17,219,220,221,222] (reviewed in [16]). Another possibility is the use of NK or NKT cells as a cellular therapy for glioblastoma [223,224]; these innate immune cells have T cell-like cytotoxic potential, but the majority of studies are at the preclinical stage. These technologies may therefore benefit from novel approaches that take into consideration the glioblastoma immune contexture.

Given that the CAR molecule features a modular design, there are many different kinds of genetic modification that can further enhance CAR-T cell function (reviewed comprehensively by others [225]), and some of these are now being considered to address the unique challenges of treating glioblastoma. A preclinical study identified IL-8 (CXCL8) as abundantly secreted from U87 glioblastoma lines after radiation, and found that enforced expression of its receptors CXCR1 or CXCR2 allows CAR-T cells to take advantage of this tumour-secreted IL-8 and home to U87-derived xenografts in mice [226]. Our review of the literature, as summarised here, suggests other chemokine pathways that may also be re-purposed to drive CAR-T cell migration and activity, such as the CCL2/CCR2 pathway. CAR-T cells with switch-signalling receptors offer a novel way to convert T cell inhibitory signals, such as those provided by PD1 [227] to activation signals such as CD28, and a similar process could potentially be used to reverse the inhibitory capacity of glioblastoma-abundant cytokines such as TGF-β, an idea which has been reviewed recently [228]. An early preclinical study of CAR-T cells with a decoy receptor for TGF-β (a non-signalling ectodomain of the TGF receptor II) prolonged the life of U87 glioblastoma bearing mice and polarized GAM towards an M1 phenotype [229].

In addition, key molecules such as IL-6, TGF-β, and CSF-1, which support the immunosuppressive, GAM-dominated microenvironment, could be targets for monoclonal antibody therapy in combination with CAR-T therapy, therapeutic vaccination, or immune checkpoint blockade. The caveat here being that the local effects of blocking such molecules in the tumour microenvironment must be separated from systemic activity of these pleiotropic signalling molecules. Again, the unique capacities of CAR-T cells may offer an avenue to tightly control expression of monoclonal antibodies or other targeted inhibitors, an approach sometimes termed a ‘biofactory’ [230] which has yet to reach its full potential and could be readily coupled to CAR-T technology. Early examples include the use of CAR-T cells to produce local IL-12 [231], which have the notable effect of invigorating an anti-tumour response from macrophages, or CAR-T cells secreting bi-specific T-cell engagers (BiTEs) to enhance tumour targeting, and this approach has recently been tested against glioblastoma in preclinical setting [232].

In conclusion, in this review we have identified a range of potential molecular targets that may allow for immune modulation of the glioblastoma microenvironment and thereby improved prospects for treatment of this tumour. Each of the molecules identified has targeted therapies in clinical testing for the treatment of other cancer types. However, with the exception of TGF-β [233,234] and CSF-1 [235], there are no clinical data thus far for their use in the treatment of glioblastoma. Galunisertib, a TGF-β inhibitor that has been tested as a monotherapy, or in combination with lomustine chemotherapy, in phase 1 and 2 trials has shown no improvement in overall survival for glioblastoma patients [233,234]. A CSF-1 receptor inhibitor, used as a monotherapy, or prior to surgical resection, also reached phase 2 trial for glioblastoma, but showed no improvement in progression free survival at 6 months [235]. A review of open clinical trials shows few active trials targeting these molecules for glioblastoma, with reports awaited for a TGF-β2-targeting antisense oligonucleotide therapy (NCT00761280), and a trial of an L-oligoribonucleotide designed to neutralize CXCL12 (NCT04121455). These registered trials will also focus on monotherapy approaches, or combination with standard surgery, chemotherapy or radiation treatments rather than testing in combination with immunotherapy. We have emphasized that the interplay between immune cells and tumour is complex, and expect that progress in immunotherapy for glioblastoma patients will depend on big data and mechanism-based studies as well as further clinical investigation.

## Figures and Tables

**Figure 1 cells-10-00607-f001:**
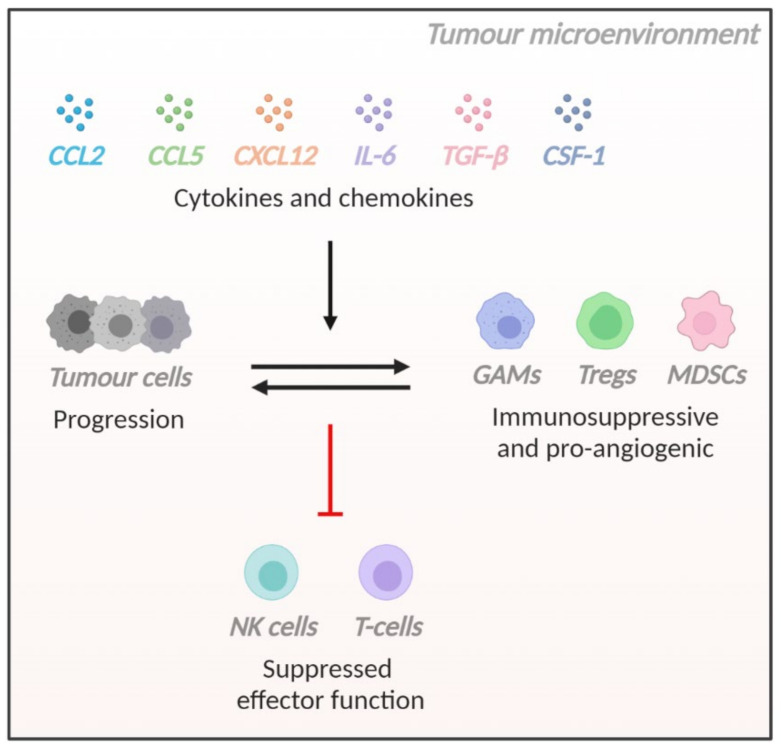
Schematic representation of the crosstalk between tumour and immune cells facilitated by key cytokines and chemokines within the glioblastoma microenvironment.

**Table 1 cells-10-00607-t001:** Key cytokines, chemokines and their respective receptors in glioblastoma.

Ligand	Alternative Name	Receptor
CCL2	MCP-1	CCR2 and CCR4
CCL5	RANTES	CCR1, CCR5 and CD44
CXCL12	PBGF or SDF-1	CXCR4 and ACKR3
IL-6	BSF-2, IFN-β2, HGF or HSF	IL-6 receptor
TGF-β	-	TGF-β receptor
CSF-1	M-CSF	CSF-1R

Monocyte chemoattractant protein-1 (MCP-1); Regulated upon Activation, Normal T cell Expressed and Secreted (RANTES); pre-B cell growth factor (PBGF); stromal cell-derived factor-1 (SDF-1); Interleukin-6 (IL-6); B-cell stimulatory factor-2 (BSF-2); interferon-β2 (IFN-β2); hybridoma growth factor (HGF); hepatocyte-stimulating factor (HSF); transforming growth factor-beta (TGF-β); colony stimulating factor-1 (CSF-1); macrophage colony-stimulating factor (M-CSF); colony stimulating factor-1 receptor (CSF-1R).

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
