# Peer review of "The Role of Cytokines and Chemokines in Shaping the Immune Microenvironment of Glioblastoma: Implications for Immunotherapy"

_cells, 2021, doi:10.3390/cells10030607_

Round 1

Reviewer 1 Report

The manuscript by Yeo et al. reviews the literature on the role of chemokines and their receptors in glioblastoma biology and therapy. Particularly the review focuses on the glioblastoma microenvironment, cell types and chemokines involved in tumor maintenance and development.  

Overall, the topic is likely to be of wide interest for publication, being both a research and clinically relevant subject, namely the role of chemokine for glioblastoma potential immunotherapy.

Major criticisms:

A concise paragraph underlying and discussing the relevance of brain immune privilege, low immunogenicity as well as immunosuppressive TME in the   immune therapy resistance of glioblastoma should be added.

A description of other therapeutic approaches than CAR-T (e.g. vaccination with peptides, adoptive T cell therapy, immune checkpoint inhibitors) would substantially complete the introduction.

Author Response

Thank you for the positive comments on our manuscript and your careful review. We address specific revision requests below:

  1. A concise paragraph underlying and discussing the relevance of brain immune privilege, low immunogenicity as well as immunosuppressive TME in the   immune therapy resistance of glioblastoma should be added.

RESPONSE: In the section on ‘The glioblastoma TME’, we have added a new paragraph at lines 126-136 that covers brain immune privilege and introduces the concept of the immunosuppressive TME. Please note that individual elements of the immunosuppressive TME (e.g. M2 macrophages, MDSC, Treg) are also described in detail in the rest of this section. In addition, we have updated the final paragraph of this section to include a brief mention of the low immunogenicity of glioblastoma, at lines 190-192.

  1. A description of other therapeutic approaches than CAR-T (e.g. vaccination with peptides, adoptive T cell therapy, immune checkpoint inhibitors) would substantially complete the introduction.

RESPONSE: We agree and have added a new section to the Introduction that briefly describes other immunotherapeutic modalities (peptide vaccine, T cell therapy, dendritic cell therapy and immune checkpoint inhibitors) tested in glioblastoma (lines 41-80 of the revised manuscript).

Reviewer 2 Report

Yeo et al review the action of six major cytokines/chemokines shaping the GBM TME.

The authors put considerable emphasis in the initial and final part of the text on CAR T cells, something that at this time looks overstated. On the other hand, nothing is said about dendritic cell (eg Mitchell et al, 2015) and T cell therapies, something that I would encourage them to modify.

Also, the role of interleukin 10 (eg Iwata et al, 2020) and IDO (eg Ladomersky et al, 2020) is neglected and this should be corrected.

Minor points

The quotation in line 41 (3% of GBM...) is somehow weird, should be rephrased.

The statement on line 60 looks contradictory with what mentioned at the beginning of the paragraph.

Author Response

Thank you for the careful review of our manuscript, and we address specific comments below:

  1. The authors put considerable emphasis in the initial and final part of the text on CAR T cells, something that at this time looks overstated. On the other hand, nothing is said about dendritic cell (eg Mitchell et al, 2015) and T cell therapies, something that I would encourage them to modify.

RESPONSE: We agree and have added a new section to the Introduction that briefly describes other immunotherapeutic modalities (peptide vaccine, T cell therapy, dendritic cell therapy and immune checkpoint inhibitors) tested in glioblastoma (lines 41-80 of the revised manuscript).

  1. Also, the role of interleukin 10 (eg Iwata et al, 2020) and IDO (eg Ladomersky et al, 2020) is neglected and this should be corrected.

RESPONSE: We have added a new section entitled ‘Additional soluble factors with immunomodulatory roles in glioblastoma’ (lines 569-605), in which we discuss the role of IL-10, as well as MIF and osteopontin. We believe that this new section helps to provide a more comprehensive overview of the diverse soluble mediators of immunosuppression in glioblastoma. In addition, a brief discussion on the role of IDO in mediating T cell immunosuppression has been added to the last paragraph of the section on ‘The glioblastoma TME’ (lines 183-187).

  1. The quotation in line 41 (3% of GBM...) is somehow weird, should be rephrased.

RESPONSE: This sentence has been rewritten (lines 71-72 of the revised manuscript) to make it fully understandable.

  1. The statement on line 60 looks contradictory with what mentioned at the beginning of the paragraph.

RESPONSE: This sentence has been rewritten so that it does not contradict the sentence at the beginning of the paragraph. Changes are now on lines 99-102 of the revised manuscript.

Reviewer 3 Report

This review is a nice summary of the cytokines/chemokines involved in GBM progression and reads easily. I have a few comments:

  1. The sentence lines 28-30 is not clear: authors should rephrase for the reader to understand that it is the first cause of cancer-related death in people aged between 15 and 34 years and not that the highest mortality rate of GBM is in the 15-34 yrs. population.
  2. The sentence lines 46-48 could be rephrased as well as the working “…has been critical to the lasting survival of…” is not totally clear.
  3. In the paragraph starting line 120, it would be interesting to cite the original studies when discussing preclinical and clinical studies instead of citing review papers.
  4. On lines 141-142, regarding the T cell infiltrate and association of CD8 T cells with survival, it is maybe worth citing the paper by Marinari et al., Oncoimmunology 2020 (https://doi.org/10.1080/2162402X.2020.1779990), which, surprisingly, finds a negative association of the T cell infiltrate with survival in GBM patients.
  5. Finally, although a few trials have been conducted in humans targeting the cited chemokines/cytokines, as mentioned by the authors in the perspectives, it would be of interest to briefly discuss the results of the published clinical trials and to briefly mention the ongoing trials.

Author Response

Thank you for the positive comments on our manuscript and your careful review. We address specific revision requests below:

  1. The sentence lines 28-30 is not clear: authors should rephrase for the reader to understand that it is the first cause of cancer-related death in people aged between 15 and 34 years and not that the highest mortality rate of GBM is in the 15-34 yrs. population.

RESPONSE: We thank the reviewer for the opportunity to improve the meaning of this sentence. Changes are on lines 31-32 of the revised manuscript.

2. The sentence lines 46-48 could be rephrased as well as the working “…has been critical to the lasting survival of…” is not totally clear.

RESPONSE: This has now been reworded – please see lines 85-88 of the revised manuscript.

3. In the paragraph starting line 120, it would be interesting to cite the original studies when discussing preclinical and clinical studies instead of citing review papers.

RESPONSE: We agree that it is best practice to cite original studies rather than reviews where practical. We have retained citations to several excellent review articles, as we believe that they collectively describe the complex roles of GAMs in the glioblastoma TME. However, we have supplemented these with two additional citations to original studies (reference 31: Markovic et al, and reference 33: Galarneau et al) describing the conflicting evidence for GAMs either promoting or inhibiting tumour growth. Please note that the clinical studies already cited in this section (Komohara et al and Zeiner et al) are original studies.  

4. On lines 141-142, regarding the T cell infiltrate and association of CD8 T cells with survival, it is maybe worth citing the paper by Marinari et al., Oncoimmunology 2020 (https://doi.org/10.1080/2162402X.2020.1779990), which, surprisingly, finds a negative association of the T cell infiltrate with survival in GBM patients.

RESPONSE: Thank you for alerting us to this interesting paper. Upon reading it, however, it seems that these authors have studied a range of glioma grades (grades II, III and IV), rather than GBM specifically. Because the biology and survival of gliomas varies so much between grades, it is difficult to draw conclusions specifically about GBM from this study. We would therefore prefer not to cite it in the context of our review specifically about GBM.

5. Finally, although a few trials have been conducted in humans targeting the cited chemokines/cytokines, as mentioned by the authors in the perspectives, it would be of interest to briefly discuss the results of the published clinical trials and to briefly mention the ongoing trials.

RESPONSE: We agree and have added some additional text to the final paragraph in the Conclusion (lines 656-666). This text covers published reports of inhibitors of TGB-beta and CSF-1, as well as a brief discussion of some open clinical trials targeting cytokines/chemokines discussed in the preceding sections.

Round 2

Reviewer 2 Report

The points raised in my first report have been properly addressed.